# Retinoic Acid Signalling in the Pineal Gland Is Conserved across Mammalian Species and Its Transcriptional Activity Is Inhibited by Melatonin

**DOI:** 10.3390/cells12020286

**Published:** 2023-01-11

**Authors:** Anna Ashton, Jason Clark, Julia Fedo, Angelo Sementilli, Yara D. Fragoso, Peter McCaffery

**Affiliations:** 1Institute of Medical Sciences, University of Aberdeen, Aberdeen AB25 2ZD, UK; 2Department of Physiopathology, Universidade Metropolitana de Santos and Centro, Universitario Lusíada, Santos 11050-071, SP, Brazil; 3Department of Post Graduate Studies, Universidade Metropolitana de Santos, Santos 11045-002, SP, Brazil

**Keywords:** retinoic acid, RAR, RALDH, Rdh10, cyp26a1, circadian, pineal gland, melatonin

## Abstract

The pineal gland is integral to the circadian timing system due to its role in nightly melatonin production. Retinoic acid (RA) is a potent regulator of gene transcription and has previously been found to exhibit diurnal changes in synthesis and signalling in the rat pineal gland. This study investigated the potential for the interaction of these two systems. PCR was used to study gene expression in mouse and human pineal glands, ex-vivo organotypic cultured rat pineal gland and cell lines. The mouse and human pineal glands were both found to express the necessary components required for RA signalling. RA influences the circadian clock in the brain, therefore the short-term effect of RA on clock gene expression was determined in ex vivo rat pineal glands but was not found to rapidly regulate *Per1*, *Per2*, *Bmal1,* or *Cry1*. The interaction between RA and melatonin was also investigated and, unexpectedly, melatonin was found to suppress the induction of gene transcription by RA. This study demonstrates that pineal expression of the RA signalling system is conserved across mammalian species. There is no short-term regulation of the circadian clock but an inhibitory effect of melatonin on RA transcriptional activity was demonstrated, suggesting that there may be functional cross-talk between these systems.

## 1. Introduction

The main function of the pineal gland is to regulate physiological rhythms by relaying the signal of circadian time to the body through the secretion of melatonin. The suprachiasmatic nucleus (SCN) of the hypothalamus, the site of the central circadian clock, drives the circadian rhythm in melatonin production in which levels peak at night and are low during the day [1]. This rhythm is present in all vertebrates, including those nocturnally active, and in all species it is mediated by a circadian rhythm in the activity of arylalkylamine *N*-acetyltransferase (AANAT), the penultimate melatonin synthetic enzyme [2]. However, the regulatory molecular mechanisms driving the nocturnal increase in AANAT activity differ between species. In non-rodent mammals, such as humans and sheep, it is regulated by post-transcriptional mechanisms only, thought to involve phosphorylation of constitutively expressed AANAT [3,4,5]. In contrast, in most rodents AANAT is also under transcriptional control with a nightly increase in *Aanat* mRNA of more than 100-fold in rats [6]. Despite its persistence across evolution, many laboratory mouse strains are deficient in melatonin [7], due to defects in one or both of the synthetic enzymes, AANAT and acetylserotonin *o*-methyltransferase (ASMT; [8,9,10]). The selective pressure for this is thought to be reproductive success due to the inhibitory effects of melatonin on testis development [10]. 

The rodent pineal gland also rhythmically expresses components of the circadian clock [11,12,13]. The master circadian clock in the SCN consists of a self-sustaining transcriptional feedback loop in which CLOCK and BMAL1 form heterodimers and bind to E-box promoter regions to activate the transcription of three Period (Per) and two Cryptochrome (Cry) genes. These encode repressor proteins which form a complex that inhibits the transcriptional activity of the CLOCK:BMAL1 complex, thereby suppressing the transcription of their own genes. This core clock machinery is also expressed in almost every mammalian cell type [14], including in the pineal gland, where it is thought to have a role in controlling the timing of *Aanat* expression [15]. The expression of most of the core clock genes in the rodent pineal gland is high during the night, with the exception of *Bmal1*, which peaks during the day [11,12,16,17]. The pineal gland is not an autonomous circadian oscillator [18], and rhythmic events are largely driven by nocturnal norepinephrine (NE) stimulation relayed from the SCN. However, some of the pineal clock gene rhythms appear to be insensitive to NE stimulation, including *Bmal1*, *Per2* and *Cry1* [16,19,20], suggesting another rhythmically active regulator may be involved. 

A potential candidate for this is the active metabolite of vitamin A, retinoic acid (RA). RA is a potent regulator of gene transcription through the activation of ligand-gated transcription factors, RA receptors (RARs; [21]). It is synthesised via a two-step oxidative pathway, which requires the enzymes retinol dehydrogenase (RDH) and retinaldehyde dehydrogenase (RALDH). RDHs are widely expressed; however, in the adult central nervous system, the expression of RALDHs is restricted to a select number of regions, which limits the localization of RA synthesis and signalling [22,23,24]. The necessary components for such synthesis and signalling have previously been shown to be present in the rat pineal gland, in which RA is produced on a diurnal basis and acts to influence *Aanat* transcription [25]. In other tissues, RA has been reported to influence clock gene expression. It can regulate E-box-dependent circadian genes such as *Per1* and *Per2* [26], inducing phase shifts in *Per2* oscillations in peripheral clocks in vivo [27,28] and entraining *Per2* rhythmic expression in vitro [29]. Furthermore, putative RA response elements (RAREs) have been identified on the regulatory regions of *Bmal1*, *Per1* and *Per2*, suggesting that they are under direct transcriptional control by RA [30,31,32]. In addition, vitamin A deficiency disrupts the daily oscillations in the expression of several clock genes in the hippocampus; it abolished the rhythm in *Per1* and phase-shifted the rhythms in *Bmal1*, *Per2*, *Cry1* and *Cry2* [30,31,32,33], suggesting a role for vitamin A metabolites in rhythmic clock gene expression. 

Previously, RA signalling has only been identified in the rat pineal gland. Given the significant species differences in this gland, the present study first determined whether the RA signalling system is also present in the human and mouse pineal gland, examining both melatonin-deficient and melatonin-proficient mouse strains. The components required for RA synthesis and signal transduction were found to be expressed in both the mouse and human pineal glands, indicating that the signalling pathway is conserved across mammalian species; however, differences between mouse strains were detected. To examine the function of RA in the pineal gland, it was then determined whether it influences the circadian clock in this gland. Although it was not found to rapidly regulate clock gene expression, examination of the interaction of RA with melatonin demonstrated that melatonin can inhibit RA-induction of gene expression, suggesting that there may be cross-talk between these two hormonal products of the pineal gland. 

## 2. Materials and Methods

### 2.1. Animals

Animals were maintained at 20–24 °C with unrestricted access to rodent chow and water. They were housed under a 12 h light:12 h dark cycle (lights on 07:00–19:00). Adult male or female C57BL/6J (C57), CBA/Ca (CBA) and MSM/Ms (MSM) mice were sacrificed by cervical dislocation at zeitgeber time (ZT) 1 or ZT7 (where ZT0 corresponds to lights on at 07:00). Pineal glands were removed and rapidly frozen on dry ice for quantitative polymerase chain reaction (qPCR) analysis. Male Sprague Dawley (SD) rats aged 6–7 weeks were sacrificed by rising CO_2_ and cervical dislocation. All animal procedures were carried out in accordance with Home Office regulations and local ethics committee guidelines.

### 2.2. Preparation of Human Tissue 

Human pineal glands from male individuals aged 35–45 years old, who did not present any neurological or psychiatric disease, were collected during necropsy procedures. Individuals whose death was related to head trauma, extensive infection, or toxic, anoxic, or metabolic injuries were excluded from the study. Samples were collected in RNAlater RNA stabilization reagent (Qiagen, Venlo, The Netherlands) within 24 h of death and stored at 4 °C for PCR analysis. 

### 2.3. Ex Vivo Pineal Gland Culture

Pineal glands from rats aged 6–7 weeks old were obtained at ZT 4-5 and were cultured based on the method used by Bailey et al. [34]. They were rapidly dissected and placed immediately into ice-cold culture medium consisting of BGJb medium (Fitton-Jackson modification; Gibco, Waltham, MA, USA) containing 1 g/L bovine serum albumin fraction V (Sigma-Aldrich, Burlington, VT, USA), 25 mM HEPES buffer (Sigma-Aldrich), 2 mM GlutaMAX supplement (Gibco), 0.1 mg/mL ascorbic acid (Stem Cell Technologies, Vancouver, BC, Canada) and 100 U/mL penicillin-streptomycin (Gibco). Meninges were removed under a dissection microscope and pineal glands were transferred onto Millicell culture plate inserts (Millipore, Burlington, VT, USA) in a 6-well plate, one pineal gland per well. Pineal glands were incubated in 1 mL culture medium at 37 °C, 5% CO_2_; media were changed after 24 and 48 h. The pineal glands were treated after 48 h in culture with RA (1 µM; Sigma-Aldrich), norepinephrine (NE; 100 nM; Sigma-Aldrich), or vehicle control for four hours. RA and NE were dissolved in dimethyl sulfoxide (DMSO; Sigma-Aldrich), therefore, control treatments received an equivalent concentration of DMSO (0.01%). Following treatment, pineal glands on the membrane inserts were rapidly frozen on dry ice for qPCR analysis.

### 2.4. Cell Culture

SH-SY5Y human neuroblastoma cells were maintained in Dulbecco’s modified eagle medium (Gibco) with 10% foetal calf serum (Gibco) and 100 U/mL penicillin-streptomycin at 37 °C, 5% CO_2_. Cells were plated in 12-well plates and treated with RA (1 µM), melatonin (100 nM; Sigma-Aldrich), or vehicle control for 24 h. RA and melatonin were dissolved in DMSO, therefore, control treatments received an equivalent concentration of DMSO (0.01%). 

### 2.5. PCR 

Total RNA was extracted from individual pineal glands using an RNeasy mini kit (Qiagen) with on-column DNase digestion (Qiagen) to remove genomic DNA. RNA was quantified using a NanoDrop spectrophotometer (Thermo Scientific, Waltham, MA, USA) and precipitated in 100% ethanol, linear acrylamide and ammonium acetate. cDNA was synthesised using a High Capacity RNA-to-cDNA kit (Applied Biosystems, Waltham, MA, USA). PCR analysis was performed using primers designed for mouse (Table 1 (A)) or human (Table 1 (B)) with an annealing temperature of 60 °C for 35 cycles. PCR products were visualised by agarose gel electrophoresis and UV transillumination. 

### 2.6. Quantitative PCR

Total RNA was extracted using an RNeasy mini kit (Qiagen) with on-column DNase digestion (Qiagen) to remove genomic DNA. RNA was quantified using a NanoDrop spectrophotometer (Thermo Scientific) and precipitated in 100% ethanol, linear acrylamide and ammonium acetate. cDNA was synthesised from 150–200 ng (mouse), 500 ng (rat and human), or 250 ng (SH-SY5Y cells) total RNA using a High Capacity RNA-to-cDNA kit (Applied Biosystems). qPCR analysis was performed using SensiMix SYBR mastermix (Bioline) using primers designed for mouse (Table 1 (A)), human (Table 1(B)), or rat (Table 1 (C)) with an annealing temperature of 60 °C for 45 cycles. Samples were run on a LightCycler 480 (Roche) and data were analysed using LightCycler 480 Software 1.5. Target gene expression was normalised to Actb (mouse) or Gapdh (rat and SH-SY5Y) expression. 

## 3. Results

### 3.1. Components Necessary for Retinoic Acid Signalling Are Expressed in the Human and Mouse Pineal Gland

For RA signalling to occur, the enzymes that catalyse the two-step oxidation of vitamin A (retinol) into RA must be present, as well as the RARs through which RA signals. PCR was performed to determine whether the genes encoding the required components are expressed in the human and mouse pineal glands. Analysis of gene expression in adult human pineal glands detected mRNA encoding the three retinaldehyde dehydrogenases (RALDHs), which catalyse the second step of RA synthesis from retinol (Figure 1a). Transcripts encoding all three RAR subtypes, RARα, RARβ and RARγ, were also detected. Similarly, analysis of gene expression in adult male CBA mouse pineal glands demonstrated that the genes encoding all three RALDHs are expressed, as well as the three RAR subtypes (Figure 1b). 

### 3.2. Retinoic Acid Receptors Are Differentially Expressed between Mouse Strains

The rhythm and magnitude of melatonin synthesis vary significantly between mouse strains and many strains are unable to produce considerable amounts of melatonin [7,8,35]. Therefore, it was determined whether there are differences in the expression of RA signalling components between strains. Expression of the genes encoding the RA synthetic enzymes and RARs were compared in the pineal glands of melatonin-proficient (MSM) and melatonin-deficient (C57) strains. The MSM strain was recently established from Japanese wild mice and retains many characteristics of wild mice [36], whereas the commonly used C57 strain has lost its ability to produce melatonin due to a mutation in the *Aanat* gene that gives rise to a truncated form of the protein [9].

Expression of the genes encoding the RA receptors (*Rara*, *Rarb*, *Rarg* and *Rorb*) and the synthetic enzymes (*Rdh10*, *Raldh1*, *Raldh2* and *Raldh3*) was determined by qPCR. Expression of *Rdh10* was comparable between strains; this encodes the enzyme that catalyses the first step in the RA synthetic pathway of the oxidation of retinol to retinaldehyde (Figure 2a). Similarly, there were no differences in the expression of the genes encoding the RALDH synthetic enzymes, which catalyse the second step of RA synthesis (Figure 2b–d). However, there was a difference in the gene encoding the RAR, *Rara*, which was higher in C57 mouse pineal glands than in MSM (Figure 2e). Expression of *Rarb* and *Rarg* also exhibited a trend towards an increase in C57 pineal glands, though this was not statistically significant (*p* = 0.1; Figure 2f,g); whereas expression of the gene encoding retinoic acid receptor-related orphan receptor (ROR) β, was comparable between mouse strains (Figure 2h). 

To determine whether the differences in the expression of RARs between mouse strains correlate with their ability to produce melatonin, another mouse strain was investigated, CBA, which is also capable of synthesising melatonin [7]. Diurnal changes in RA signalling genes were previously observed in the rat pineal gland [25], therefore, gene expression at two time points was determined to exclude effects of the time of day. MSM, C57 and CBA mouse pineal glands were collected at ZT1 and ZT7. Expression of the genes encoding a RAR, *Rara*, and enzymes catalysing the first and second synthetic steps, *Rdh10* and *Raldh1*, respectively, was determined by qPCR. Expression of *Aanat* was also determined as a positive control.

As expected, there was an effect of both time (F1, 11 = 30.96, *p* = 0.0002) and mouse strain (F2, 11 = 6.372, *p* = 0.0145) on *Aanat* expression, as well as a time and strain interaction effect (F2, 11 = 5.13, *p* = 0.0267). *Aanat* expression was higher at ZT1 compared to ZT7 in both melatonin-proficient strains, MSM and CBA (*p* < 0.01), but exhibited low expression at both time points in the melatonin-deficient strain, C57 (Figure 3a). In line with this, *Aanat* was higher in MSM and CBA than in C57 pineal glands at ZT1 only. *Aanat* expression peaks during the night and, at ZT1, shortly after the start of the light period, it had not returned to the baseline day-time levels in the melatonin-proficient strains [10]. There were no significant changes in the expression of *Rdh10* between time points (F1, 11 = 0.9114, *p* = 0.3603). Overall, there was an effect of strain on *Rdh10* (F2, 11 = 4.921, *p* = 0.0298; Figure 3b); however, post hoc comparisons did not identify any significant changes between individual strains, although expression appears to be higher in CBA mice compared to MSM (*p* = 0.0688, for ZT1). Differences were observed in *Raldh1* expression between strains (F2, 11 = 12.9, *p* = 0.0013), with higher levels in CBA pineal glands than in both MSM and C57 at ZT1 (Figure 3c); a similar trend was also observed at ZT7 though this was not statistically significant. However, *Raldh1* expression remained comparable between the two time points in each of the mouse strains (F1, 11 = 0.431, *p* = 0.5250). There was an effect of mouse strain on *Rara* expression (F2, 11 = 124.1, *p* < 0.0001), and a strain and time interaction effect (F2, 11 = 6.024, *p* = 0.0171). Expression of *Rara* was low in MSM pineal glands at both time points, with higher expression observed in both C57 and CBA pineal glands (Figure 3d). Expression was higher still in CBA compared to C57, at ZT1 only. In line with this, there was a significant difference in *Rara* expression between the two time points in the C57 pineal glands, with higher expression at ZT7 compared to ZT1 (*p* < 0.05). However, there were no changes between time points in the other mouse strains (F1, 11 = 0.9227, *p* = 0.3574). 

Overall, these results demonstrate that *Rara*, *Rdh10* and *Raldh1* are differentially expressed between mouse strains, with generally the highest expression in CBA pineal glands and the lowest in MSM. 

### 3.3. Retinoic Acid and Pineal Clock Gene Expression

RA synthesis has previously been shown to exhibit diurnal changes in the rat pineal gland [25], which may enable it to serve a role in the regulation of molecular rhythms here, such as oscillating clock gene expression. In addition, RA has previously been shown to influence the expression of clock genes in other tissues, including *Per1* and *Per2* [26,27]. It was therefore investigated whether RA regulates the expression of clock genes in the rat pineal gland.

The majority of rhythmically expressed genes in the pineal gland are regulated by the nocturnal activation of adrenergic receptors by NE [34], including the clock gene *Per1* [16,20]. Therefore, the effect of RA on clock gene expression in the presence of NE was also determined in order to replicate the in vivo environment in which RA would be active. 

Cultured rat pineal glands were treated with vehicle control, 1 µM RA, 100 nM NE, or RA combined with NE, for four hours, and the expression of *Per1*, *Per2*, *Cry1* and *Bmal1* was determined by qPCR. Expression of *Rarb* was also determined as a positive control, which is induced by RA through a RARE on its promoter [37]. There were no changes in *Per1* (F3, 8 = 2.436, *p* = 0.1397) or *Cry1* (F3, 8 = 0.6209, *p* = 0.6210) in response to any of the treatments (Figure 4a,c). Overall, there were differences in *Per2* expression between treatments (F3, 8 = 4.373, *p* = 0.0423). However, post hoc comparisons did not identify any significant differences between individual treatments, although there was a trend towards an increase in *Per2* in response to NE (*p* = 0.0549; Figure 4b). Overall, there were differences in *Bmal1* expression between treatments (F3, 8 = 7.96, *p* = 0.0087); *Bmal1* was upregulated by NE by 1.4-fold (Figure 4d), yet it did not appear to change in response to NE combined with RA. However, the difference in *Bmal1* expression between NE alone and NE combined with RA was not statistically significant. Treatment of RA alone also did not influence *Bmal1* expression. However as expected, there were significant changes in *Rarb* expression (F3, 8 = 16.56, *p* = 0.0009), with more than a four-fold increase in *Rarb* in response to both RA alone and NE combined with RA (Figure 4e), indicating that RA was active. 

### 3.4. Melatonin Inhibits Induction of Gene Expression by Retinoic Acid

RA synthesis in the pineal gland increases during the night, when melatonin production is also switched on [25]. Synchronised synthesis of RA and melatonin by the pineal gland in vivo may mean RA and melatonin act together to influence gene expression. It has been reported that melatonin and RA synergise to inhibit tumour development [38,39] and previous studies suggest that melatonin can potentiate the induction of gene expression by RA [40,41]. Therefore, it was determined whether melatonin affects RA-induced gene expression by examining the expression of two established RA-responsive genes, *CYP26A1* and *RARB* [37,42]. SH-SY5Y neuroblastoma cells were treated with 1 µM RA, 100 nM melatonin, RA combined with melatonin or vehicle control for 24 h, and gene expression was determined by qPCR. There were significant differences in *CYP26A1* between the treatments (F3, 8 = 44.60, *p* < 0.0001); as expected, RA induced upregulation of *CYP26A1* with a 9-fold increase compared to control (Figure 5a). Treatment with RA combined with melatonin also induced upregulation of *CYP26A1*; however, this was lower than the effect of RA alone with an increase of only 6-fold. This suggests that the RA-induced expression of *CYP26A1* is inhibited by melatonin. Melatonin alone did not affect the expression of *CYP26A1*; in line with this there was a significant difference between RA and melatonin treatments (*p* < 0.001). The same effect was observed with *RARB* expression (F3, 8 = 91.97, *p* < 0.0001); RA treatment induced a 4.8-fold increase in *RARB* mRNA (Figure 5b), while melatonin significantly reduced induction by RA to 3.5-fold. Melatonin alone did not affect the expression of *RARB*, and there was a significant difference between RA and melatonin treatments (*p* < 0.001). 

## 4. Discussion

This study has demonstrated that the genes required for RA signalling are expressed in both the mouse and human pineal glands, with differences in the expression of RA synthetic enzymes and RARs observed between mouse strains. In the conditions examined here, RA was not found to influence clock gene expression in the rat pineal gland following a short-term treatment in vitro. However, the transcriptional activity of RA was found to be inhibited by melatonin, the nocturnal product of this gland, suggesting that it modulates the effects of RA. 

RA was previously found to be synthesised and active in the rat pineal gland [25]. The present study demonstrates that the genes encoding the necessary components for RA synthesis and signalling are also expressed in the mouse and human pineal glands, indicating that they both have the potential to synthesise and respond to RA. The conservation of this signalling pathway across mammalian species suggests that it serves an important role in this gland. The key difference between rodent and human pineal glands is the molecular mechanism that drives the rhythm in AANAT activity and subsequent melatonin synthesis. In rodents, it is under both transcriptional and post-translational control, whereby phosphorylation of AANAT increases substrate binding and protects against proteasomal degradation [4,43,44]. In contrast, in humans, while the exact mechanism is poorly understood, it is exclusively under post-translational control with stable *Aanat* expression throughout the day and night [5,45]. Therefore, if RA is involved in the regulation of the AANAT rhythm through a conserved mechanism in the human and rodent pineal gland, it is expected to be through post-translational regulation. Indeed, in addition to its classical transcriptional activities, RA also has non-genomic effects including regulation of protein translation [46] and kinase phosphorylation [47,48], and RA was previously found to regulate ERK phosphorylation in the rat pineal gland, yet it did not have a short-term effect on *Aanat* transcription [25]. However, further work is required to determine whether RA functions through non-genomic pathways in vivo.

In the present study, differences in the expression of genes encoding the RARs and RA synthetic enzymes were observed between mouse strains. Initially, the lower levels in the melatonin-proficient MSM strain were thought to be due to an inhibitory effect of melatonin on RA signalling gene expression. This has previously been reported to occur in the rodent hypothalamus; expression of RARs and RA synthetic enzymes are reduced under short photoperiod when melatonin production is high [49,50]. Furthermore, melatonin administration inhibits the expression of *Raldh1*, as well as *Cyp26b1*, which encodes a RA catabolic enzyme, and *Stra6*, encoding a retinol transporter [50]. Melatonin is therefore thought to dampen overall RA signalling. However, on further examination, comparison with another melatonin-proficient strain suggested that the expression levels of RA signalling genes in the pineal gland do not correspond with the ability to produce melatonin; of the three strains studied, expression was generally lowest and highest in the two melatonin-proficient strains, MSM and CBA, respectively, with expression in the melatonin-deficient strain, C57, between the two.

The lack of correlation between RA signalling gene expression and melatonin proficiency may indicate that RA is not involved in melatonin synthesis. However, while the C57 pineal gland is generally considered to be melatonin-deficient due to the truncated AANAT protein, it is capable of producing melatonin and retains the necessary molecular machinery for its rhythmic synthesis, including the transcriptional regulators of AANAT [51]. Furthermore**,** studies using short sampling intervals have demonstrated that there is a brief nocturnal peak in melatonin [35,52]. Therefore**,** C57 mice are not completely melatonin-deficient and it is likely a more complex comparison than melatonin deficiency versus proficiency. The melatonin peak in the C57 pineal gland occurs in the middle of the night, whereas, in CBA and MSM strains, melatonin peaks towards the end of the night [7,10], suggesting there are variations in the timing of pineal rhythms between strains. This may account for the differences observed in RA signalling gene expression in this study if the genes are subject to diurnal changes but display different rhythms. Indeed, *Rara* was found to change between ZT1 and ZT7 in C57 pineal glands only. There were no other diurnal changes observed but only two time points were studied, both of which occurred during the light period. Further studies examining more frequent time points across the light/dark cycle are required to determine whether there is a rhythm in RA signalling gene expression in mice and, if so, whether it varies between strains. At present, the reasons for the differences in RA signalling gene expression between mouse strains remain unclear; whether they have a physiological purpose or are a downstream effect of variations in rhythmic melatonin synthesis is yet to be determined. Further examination of the relevant differences between strains may provide insight into the role of RA in the pineal gland and the mechanisms which regulate its activity. 

While the expression levels of RA signalling genes did not appear to correspond with melatonin levels, the data indicate that melatonin can reduce RA-induced gene expression. This suggests that there is an interaction between RA and melatonin signalling. This result was unexpected as previous studies have reported synergistic actions of melatonin and RA on tumour development [38,39] and apoptosis [53], implying a potentiation of RA activity by melatonin. Indeed, melatonin potentiates RA-induction of cone arrestin (CAR) in retinoblastoma cells, through enhanced activity at the CAR gene promoter [40]. Furthermore, melatonin has been shown to enhance the transcriptional activity of RARα [41,53,54] and its binding to RAREs [41] in breast cancer cells. This effect of melatonin is thought to be mediated by G proteins coupled to the melatonin receptor, MT1 [54]. Interestingly, melatonin also suppresses the transcriptional activity of other nuclear receptors, including the oestrogen and glucocorticoid receptors [41,55], through activation of the same receptor but via a different set of G proteins [54]; in MCF-7 cells, Gαi2 was found to mediate the modulation of oestrogen receptor (ER) α signalling by melatonin, whereas Gαq and Gα11 proteins mediated the modulation of RARα signalling. This indicates that melatonin has differential effects on the transcriptional activity of hormone receptors through the activation of the same receptor in the same cell type. 

Melatonin is also thought to act via the nuclear receptor, RORα [56,57,58], and studies have suggested this receptor may also be involved in the synergistic effects of RA and melatonin. RA upregulates *Rora* expression in retinoblastoma cells [40], while putative ROR response elements (ROREs) have been identified on promoter regions of *Rara* and *Rarb* in human and rat [30,59], suggesting that RA and RORα enhance each other’s activity. Transcriptional cross-talk has also been shown to occur between RORα and RAR; however, this can be either repression or enhancement of RORα activation by RAR depending on RA availability [60].

It is unclear why the effects of melatonin on RA activity observed here are inconsistent with previous studies. It is possible that melatonin acts through different mechanisms to inhibit or enhance RA transcriptional activity and so the effects are dependent on cell type. Indeed, RA-induction of transcription is cell type-specific [61,62] and the present experiments were performed in a neuronal cell line, whereas previous studies predominantly studied breast cancer cells. The effects of melatonin may also be gene-specific and dependent on the mechanism of RA-induced transcription. The genes investigated here, *Cyp26a1* and *Rarb*, are directly regulated by RA with well-defined RAREs in their promoter regions [37,42], but RA can also act through indirect mechanisms to regulate gene transcription [63]. However, as melatonin was previously shown to enhance the activation of a RARE-luciferase reporter construct by RA [41], this is not likely to be the reason for the different effects. The present study only investigated genes involved in RA signalling, which have previously been reported to be inhibited by melatonin in the hypothalamus, as described above. Therefore, it is possible that, generally, melatonin enhances the transcriptional activity of RA, as previous studies have demonstrated, and the inhibitory effect is limited to genes involved in the signalling of RA itself. 

Nevertheless, it is important to understand the relationship between melatonin and RA signalling for their clinical applications; their combined use has been proposed for cancer treatment to reduce RA dose in order to alleviate toxic side effects without affecting clinical efficacy [38,39]. Individually they have also been proposed as potential treatments for Alzheimer’s disease [64,65]; therefore, it is worth determining whether their combined application grants greater therapeutic benefit. However, the results of the present study indicate that their relationship is complex and involves both positive and negative interactions, which are potentially dependent on cell type. 

The conservation of RA signalling components across mammalian species suggests that RA may play an essential role in the pineal gland. We investigated whether RA regulates clock gene expression in the rat pineal gland. However, the evidence presented here suggests that RA does not rapidly regulate clock gene expression in this region, with or without adrenergic stimulation present. RA regulates the expression of E-box-dependent circadian genes such as *Per1* and *Per2* in other tissues [26,27] and putative RAREs have been identified on the promoter regions of *Per1*, *Per2* and *Bmal1* [30,31,32], yet RA was not found to influence their expression here. In the present study, the short-term effects of RA were examined using a treatment period of four hours, as this is relevant to circadian rhythms. However, RA was previously found to enhance *Per1* and *Per2* expression following a longer treatment of 16 h [26]; therefore, the length of the treatment may account for the lack of an effect. Therefore, a time-course experiment should be performed in future studies before a role for RA in clock gene regulation in the pineal gland can be dismissed. In addition, and somewhat unexpectedly, *Per1* expression did not respond to NE treatment, despite being reported to be under adrenergic control [20,66]. However, in previous studies it was rapidly induced, reaching peak levels after just one or two hours of stimulation; therefore, it is likely that the four-hour treatment performed here was too long to see a significant effect. An increase in *Bmal1* in response to NE was observed here though, which has not previously been reported. However, analysis of its rhythmic expression in the pineal gland in vivo has demonstrated that it increases gradually during the night [20], suggesting that the nocturnal release of NE may play a role in its induction. 

## 5. Conclusions

In conclusion, this study has demonstrated that the genes required for RA synthesis and signalling are expressed in the human and mouse pineal glands, suggesting that RA plays an important role in this gland as it is conserved across mammalian species. Further work is required to elucidate the nature of this role; RA can influence the transcription of more than 500 protein-coding genes [63] and a significant number of non-coding genes [67]; therefore, it may constitute a central transcriptional regulator in the pineal gland with multiple gene targets. Interestingly, this study also identified a novel inhibitory effect of melatonin, the other pineal hormone, on RA transcriptional activity. This demonstrates that the relationship between these hormones is more complex than the solely synergistic interactions previously reported. Melatonin is a well-established hormonal signal of time, and RA is emerging as a key player in biological timing. Further investigation of their interaction will be important to understand the role of RA in the pineal gland and circadian rhythms, as well as being valuable for the clinical applications of their combined use. 

## Figures and Tables

**Figure 1 cells-12-00286-f001:**
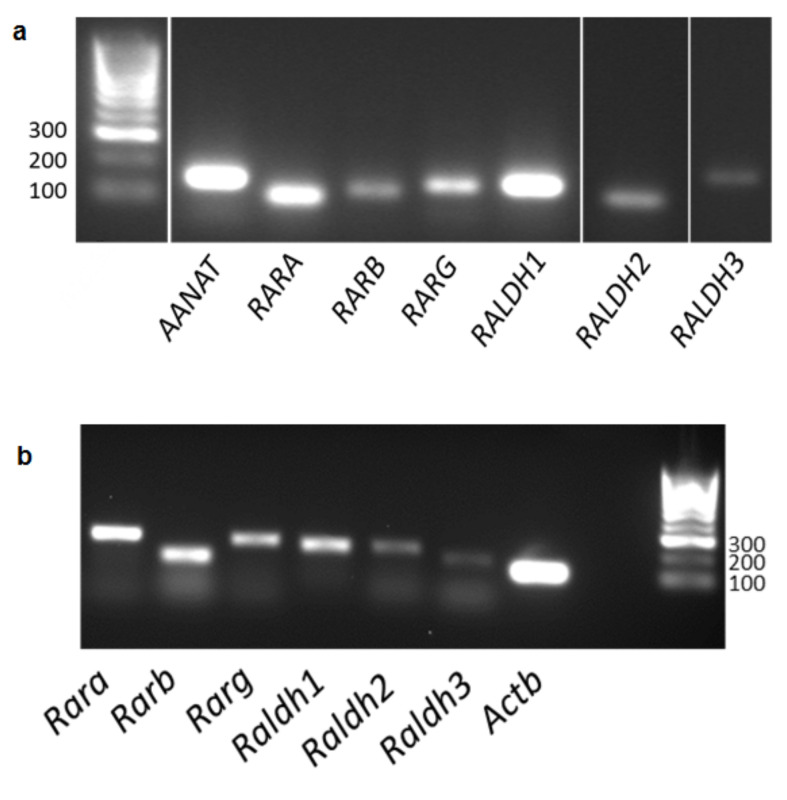
Components necessary for retinoic acid signalling are expressed in the human and mouse pineal glands. Expression of transcripts encoding the retinoic acid receptors, RARα, RARβ and RARγ, and the three retinaldehyde dehydrogenase synthetic enzymes, RALDH1, RALDH2 and RALDH3, in the adult male human pineal gland (**a**) and adult male CBA mouse pineal gland (**b**). Expression of AANAT was also determined as a positive control in the human pineal gland. Determined by PCR and gel electrophoresis.

**Figure 2 cells-12-00286-f002:**
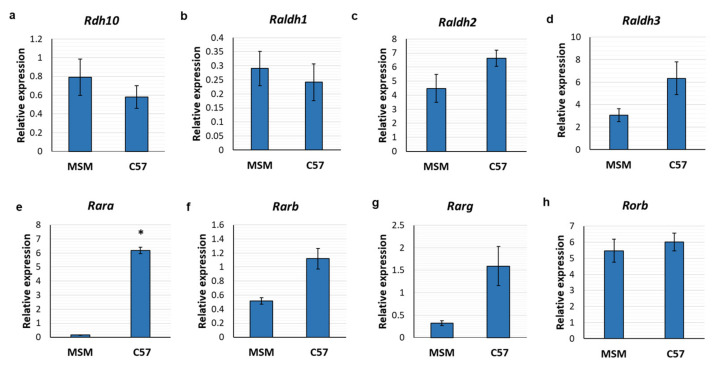
Retinoic acid receptors are differentially expressed between mouse strains. qPCR analysis of the expression of genes encoding the RA synthetic enzymes (**a**–**d**) and RA receptors (**e**–**h**) in MSM and C57 mouse pineal glands collected at zeitgeber time 1. Values represent mean mRNA expression relative to Actb, ±SEM. *n* = 3–5 pineal glands. * *p* < 0.05; determined by Mann-Whitney test.

**Figure 3 cells-12-00286-f003:**
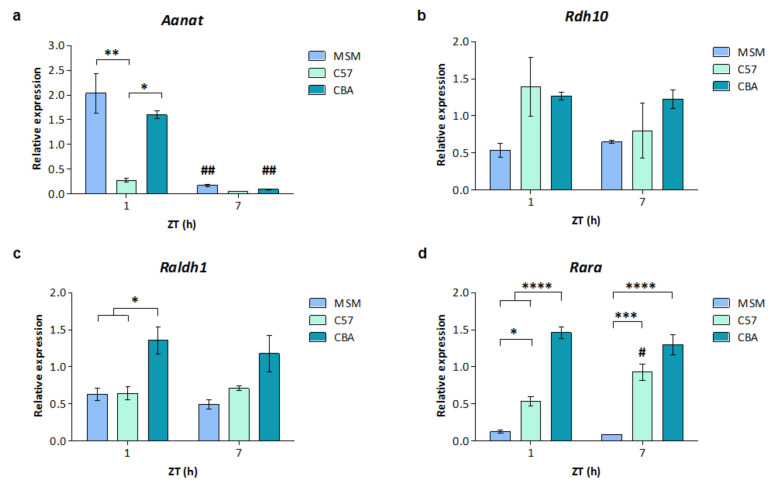
Strain differences in *Rara*, *Rdh10* and *Raldh1* expression. qPCR analysis of the expression of genes encoding the melatonin synthetic enzyme *Aanat* (**a**), RA synthetic enzymes (**b**,**c**) and the RA receptor, *Rara* (**d**), in MSM, C57 and CBA mouse pineal glands collected at zeitgeber time (ZT) 1 and ZT7. Values represent mean mRNA expression relative to *Actb*, ± SEM. *n* = 3–4 pineal glands, with the exception of C57 and MSM ZT7 groups where *n* = 2. * *p* < 0.05; ** *p* < 0.01, *** *p* < 0.001; **** *p* < 0.0001. # *p* < 0.05; ## *p* < 0.01, compared to ZT1 of the same strain. *p* values indicate post-hoc comparisons, with the exception of Rdh10 which indicates the overall strain effect. Statistical significance was determined by two-way ANOVA with Bonferroni’s multiple comparisons test.

**Figure 4 cells-12-00286-f004:**
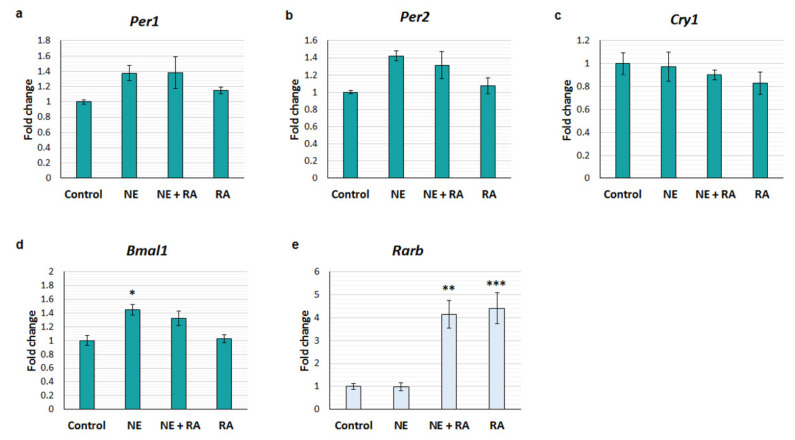
Retinoic acid does not rapidly influence clock gene expression in adult cultured pineal glands. qPCR analysis of the expression of clock genes (**a**–**d**) and the RA receptor gene, *Rarb* (**e**), in cultured pineal glands following 4 h treatment with vehicle control, 100 nM norepinephrine (NE), 1 µM retinoic acid (RA), or NE and RA combined (NE + RA). Values represent the fold change in mean mRNA expression compared to control, ± SEM. *n* = 3 glands per treatment. * *p* < 0.05; ** *p* < 0.01; *** *p* < 0.001, compared to control treatment; determined by one-way ANOVA with Tukey’s multiple comparison test.

**Figure 5 cells-12-00286-f005:**
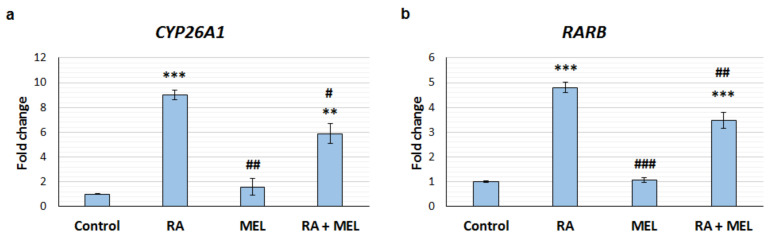
Melatonin inhibits the induction of gene expression by retinoic acid. qPCR analysis of the expression of RA-responsive genes, *CYP26A1* (**a**) and *RARB* (**b**), in SH-SY5Y cells following 24 h treatment with vehicle control, 1 µM retinoic acid (RA), 100 nM melatonin (MEL), or RA and MEL combined (RA + MEL). Values represent the fold change in mean mRNA expression compared to control, ± SEM. N = 3. *CYP26A1*: ** *p* < 0.01, *** *p* < 0.001, compared to control treatment; # *p* < 0.05, compared to RA treatment; ## *p* < 0.01, compared to RA + MEL treatment. *RARB*: *** *p* < 0.001, compared to control treatment; ## *p* < 0.01, compared to RA treatment; ### *p* < 0.001, compared to RA + MEL treatment. Determined by one-way ANOVA with Tukey’s multiple comparison test.

**Table 1 cells-12-00286-t001:** Sequences of primers used for mouse (A), human/SH-SY5Y cells (B) and rat (C).

	Gene	RefSeq Code	Product Size (bp)	Forward Primer (5’-3’)	Reverse Primer (5’-3’)
A	*Raldh1*	NM_013467.3	196	ACGTGGAAGAAGGGGACAAGGCTG	GCAAAGACTTTCCCACCATTGAGTGCC
	*Raldh2*	NM_009022.4	198	CAAGGAGGCTGGCTTTCCACCC	GGGCTCTTCCCTCCGAGTTCCA
	*Raldh3*	NM_053080.3	150	TCAAAGAGGTCGGGTTCCCTCCG	AGGCGGCTTCTCTGACCAGCT
	*Rara*	NM_009024.2	247	GAGGGCTGTAAGGGCTTCTTCCG	TGAGCTCGCCCACCTCAGGC
	*Rarb*	NM_011243.1	134	ACACCACGAATTCCAGCGCTGAC	CAGACCTGTGAAGCCCGGCA
	*Rarg*	NM_011244.4, NM_001042727.2	218	CCTGTGAAGGCTGCAAGGGCT	GTCGGGCGAGCCCTCCTCTT
	*Rdh10*	NM_133832.3	241	GCTGGAGTTGAGGATTACTGTGCCAG	GCTGGTCAGTGAGGATGGCCC
	*Rorb*	NM_001043354.2	136	CGGCCACATCATGCGAGCACA	GGCATTGTTCTGCTGGCTCCTCC
	*Aanat*	NM_009591.3	131	TCCGGCACTTCCTAACCCTGTGT	CCTGTGTAGTGTCAGCGACTCCTGA
	*Actb*	NM_007393.5	112	CCACACCCGCCACCAGTTCG	TACAGCCCGGGGAGCATCGT
B	*Raldh1*	NM_013467.3	196	ACGTGGAAGAAGGGGACAAGGCTG	GCAAAGACTTTCCCACCATTGAGTGCC
	*Raldh2*	NM_009022.4	198	CAAGGAGGCTGGCTTTCCACCC	GGGCTCTTCCCTCCGAGTTCCA
	*Raldh3*	NM_053080.3	150	TCAAAGAGGTCGGGTTCCCTCCG	AGGCGGCTTCTCTGACCAGCT
	*Rara*	NM_009024.2	247	GAGGGCTGTAAGGGCTTCTTCCG	TGAGCTCGCCCACCTCAGGC
	*Rarb*	NM_011243.1	134	ACACCACGAATTCCAGCGCTGAC	CAGACCTGTGAAGCCCGGCA
	*Rarg*	NM_011244.4, NM_001042727.2	218	CCTGTGAAGGCTGCAAGGGCT	GTCGGGCGAGCCCTCCTCTT
	*Rdh10*	NM_133832.3	241	GCTGGAGTTGAGGATTACTGTGCCAG	GCTGGTCAGTGAGGATGGCCC
	*Rorb*	NM_001043354.2	136	CGGCCACATCATGCGAGCACA	GGCATTGTTCTGCTGGCTCCTCC
	*Aanat*	NM_009591.3	131	TCCGGCACTTCCTAACCCTGTGT	CCTGTGTAGTGTCAGCGACTCCTGA
	*Actb*	NM_007393.5	112	CCACACCCGCCACCAGTTCG	TACAGCCCGGGGAGCATCGT
C	*Per1*	NM_001034125.1	137	CCAGTGGTGGGAGGCACCCT	ATGATGTCCGACTCCGGGGGC
	*Per2*	NM_031678.1	105	AAAACTGCTCCACGGGGCGG	CGTCAGGGCTGGGGTGAGTG
	*Cry1*	NM_198750.2	81	CTGACCCGCGGCGACCTATG	GCTCCAGTCGGCGTCAAGCA
	*Bmal1*	NM_024362.2	135	CGGGCGACTGCACTCACACA	GCCAAAATAGCCGTCGCCCTCT
	*Gapdh*	NM_017008.4	119	GGGCTCTCTGCTCCTCCCTGT	CAGGCGTCCGATACGGCCAAA

## Data Availability

All data are contained within the article or supplementary material. Raw data are available on request.

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
