# Peer review of "Retinoic Acid Signalling in the Pineal Gland Is Conserved across Mammalian Species and Its Transcriptional Activity Is Inhibited by Melatonin"

_cells, 2023, doi:10.3390/cells12020286_

Round 1
Reviewer 1 Report
Retinoic acid (RA) signaling activity was previously found to exhibit diurnal changes in the pineal gland. This manuscript provides important new information on the interaction of RA and melatonin in the pineal gland. The authors found that human and mouse pineal gland both express components necessary for RA synthesis and signaling. Alteration of RA signaling was not found to regulate the circadian clock, but melatonin was found to suppress activation of genes normally regulated by RA signaling. Overall, these studies have advanced our knowledge of factors regulating the circadian clock.
Specific Points to Address:
1. Introduction - In the third paragraph of the Introduction it would be good to add a general reference describing the components of the RA signaling system such as the review by Ghyselinck and Duester (Development 2019).
2. Discussion - At the end of the second paragraph of the Discussion it would be useful to indicate that more work needs to be done to determine if RA can indeed function non-genomically in vivo and whether this mechanism is required for certain processes such as the circadian rhythm.
Author Response
- Introduction - In the third paragraph of the Introduction it would be good to add a general reference describing the components of the RA signaling system such as the review by Ghyselinck and Duester (Development 2019).
- This reference has been added on page 2, line 70.
- Discussion - At the end of the second paragraph of the Discussion it would be useful to indicate that more work needs to be done to determine if RA can indeed function non-genomically in vivo and whether this mechanism is required for certain processes such as the circadian rhythm.
- This has been added on page 10, lines 427-428.
Reviewer 2 Report
Thank you for inviting me to review the submission titled by “Retinoic acid signalling in the pineal gland is conserved across mammalian species and its transcriptional activity is inhibited by melatonin”.
In this study, Anna Ashton et al detected the RA signaling expression in pineal glands, cultured pineal gland and cell lines, and concluded that the expression of the RA signaling system is conserved across mammalian species. They found RA does not involve short-term regulation of the circadian clock, but there is functional cross-talk between this system and melatonin, consisting of both stimulatory and inhibitory interactions.
My major concern is this study lacks of a solid result to support the conclusion. For example, the data verified at the level of protein expression, such as using immunofluorescence staining or Western blot results, only examining gene expression at mRNA level is very limited.
The authors assert functional cross-talk between the RA signal pathway and melatonin, but there isn’t any experiment to directly support their interactions.
The function of RA signals pathway is important, and it has been well proved that RA is extensively expressed during development and adulthood in different animal models. So what will the result “RA signaling system is conserved across mammalian species” increment to the current knowledge?
The results section is supposed to be objective. The interpretation of the results should be in the discussion section. The sample size of each experiment should be reported.
Author Response
- My major concern is this study lacks of a solid result to support the conclusion. For example, the data verified at the level of protein expression, such as using immunofluorescence staining or Western blot results, only examining gene expression at mRNA level is very limited.
- We have previously shown in rat that the transcript expression is representative of the corresponding protein. qPCR analysis allowed the comparison of multiple species without the need of antibodies that could cross species to detect the corresponding proteins; such antibodies are not always reliable. Regarding the data on the effects of RA and melatonin, we aimed to examine the transcriptional effects of RA specifically because the most widely accepted function of RA via RAR is as a transcription factor (genomic action) and state our conclusions appropriately.
- The authors assert functional cross-talk between the RA signal pathway and melatonin, but there isn’t any experiment to directly support their interactions.
- Agreed. We have edited the text on page 1, lines 23-25 and page 2, lines 97-98.
- The function of RA signals pathway is important, and it has been well proved that RA is extensively expressed during development and adulthood in different animal models. So what will the result “RA signaling system is conserved across mammalian species” increment to the current knowledge?
- We meant that this was conserved in the pineal gland specifically with the implication that this is important for pineal gland function because we have shown that RA presence in the pineal gland is not unique to just one species. In the adult central nervous system, the number of regions expressing RA synthetic enzymes is greatly reduced compared to the developing embryo, so the presence of synthetic enzymes in the pineal gland suggests an important functional role.
- The results section is supposed to be objective. The interpretation of the results should be in the discussion section. The sample size of each experiment should be reported.
- Interpretation has been moved to the discussion section.
- Sample sizes for each experiment are reported in the figure legends.
Reviewer 3 Report
The manuscript was written by Aston et al. entitled “Retinoic acid signalling in the pineal gland is conserved across mammalian species and its transcriptional activity is inhibited
by melatonin”
The analysis done is detailed and well-organized, which gives plenty of its merits. However, the discussion is a bit poor, the authors should elaborate on the discussion of the results obtained. Additionally, the main problem with this article lies in the experimental design and statistical analysis.
The appropriate reasons for reducing the experimental samples are not mentioned in detail. The assumptions for ANOVA should be verified before running the ANOVA test. The assumptions of normality and normal variance would be very difficult to test with small sample sizes.
Common comments are as follows:
The English needs to be improved and please double-check grammar and spelling.
The introduction section should be properly revised.
Significant (p-values) should be prepared according to the statistical analysis.
Whether genomic DNA was removed from RNA before cDNA synthesis (Page7, line140).
What is the annealing temperature in PCR and how many cycles
The authors give the gene names for the tested mRNA, oct4 is to my knowledge now called pou5f1.
Author mentioned for control gene expression was studied Actb (mouse) or Gapdh (rat and SH-SY5Y) expression, but I could not find it, please add this data.
Conclusions may be more appealing if authors include key insights and future prospects, rather than just summarizing the results.
Since many abbreviations are mentioned in the manuscript, the authors can add a separate Abbreviation section for readers' understanding.
Author Response
- The analysis done is detailed and well-organized, which gives plenty of its merits. However, the discussion is a bit poor, the authors should elaborate on the discussion of the results obtained. Additionally, the main problem with this article lies in the experimental design and statistical analysis.
- we have expanded the discussion of the results throughout the discussion section but in particular page 11, lines 463-468 and 505-506 as well as page 12, lines 510-513, 522-526 and 547-555.
- The appropriate reasons for reducing the experimental samples are not mentioned in detail. The assumptions for ANOVA should be verified before running the ANOVA test. The assumptions of normality and normal variance would be very difficult to test with small sample sizes.
- The sample size is low in some cases because we were limited by the number of animals from which pineal glands could be obtained.
- We agree that it can be difficult to determine normality in small data sets. We have re-analysed the data in figure 2 using a Mann-Whitney test which does not assume Gaussian distribution. Regarding the data analysed by ANOVA, we assessed these data and it does meet the criteria of being normally distributed in that the mean and median of each group are similar, and the standard deviation is less than half the mean value. Also the ANOVA is robust against deviations from normally distributed data.
Common comments are as follows:
- The English needs to be improved and please double-check grammar and spelling.
- Improvements have been made throughout the text, and grammar and spelling have been checked.
- The introduction section should be properly revised.
- We have revised the introduction section, in particular page 2, lines 68-77
- Significant (p-values) should be prepared according to the statistical analysis.
- p-values were determined and are included in the text where appropriate.
- Whether genomic DNA was removed from RNA before cDNA synthesis (Page 7, line140).
- We performed on-column DNase digestion during the RNA extraction to remove genomic DNA, we have clarified this on page 3, lines 166.
- What is the annealing temperature in PCR and how many cycles
- This information has been added to page 4, lines 171-172.
- The authors give the gene names for the tested mRNA, oct4 is to my knowledge now called pou5f1.
- We did not study Oct4 or mention this in the manuscript.
- Author mentioned for control gene expression was studied Actb (mouse) or Gapdh (rat and SH-SY5Y) expression, but I could not find it, please add this data.
- Actb and Gapdh expression were analysed as reference genes for qPCR and the expression of target genes (such as Rdh10, Raldh1) were normalised to this, as mentioned in the methods section (page 4, lines 187-188). The normalised gene expression data are shown in figures 2-5.
- Conclusions may be more appealing if authors include key insights and future prospects, rather than just summarizing the results.
- We have expanded the conclusion to include this in particular page 10, lines 461-464, page 11, lines 467-468 and lines 505-510, page 12, lines 522-526, 533-535, 547-552 and 554-555.
- Since many abbreviations are mentioned in the manuscript, the authors can add a separate Abbreviation section for readers' understanding.
- We added a table of abbreviations used in the manuscript.
Round 2
Reviewer 3 Report
Well improved, Kindly accept it
Author Response
Thank you very much for your useful suggestions!